# Accurate Identification of Spatial Domain by Incorporating Global Spatial Proximity and Local Expression Proximity

**DOI:** 10.3390/biom14060674

**Published:** 2024-06-09

**Authors:** Yuanyuan Yu, Yao He, Zhi Xie

**Affiliations:** 1State Key Laboratory of Ophthalmology, Zhongshan Ophthalmic Center, Sun Yat-sen University, Guangzhou 510060, China; weiyy29@mail2.sysu.edu.cn; 2Center for Precision Medicine, Sun Yat-sen University, Guangzhou 510080, China

**Keywords:** spatial transcriptomics, spatial domain identification, spatial embedding, graph attention network

## Abstract

Accurate identification of spatial domains is essential in the analysis of spatial transcriptomics data in order to elucidate tissue microenvironments and biological functions. However, existing methods only perform domain segmentation based on local or global spatial relationships between spots, resulting in an underutilization of spatial information. To this end, we propose SECE, a deep learning-based method that captures both local and global relationships among spots and aggregates their information using expression similarity and spatial similarity. We benchmarked SECE against eight state-of-the-art methods on six real spatial transcriptomics datasets spanning four different platforms. SECE consistently outperformed other methods in spatial domain identification accuracy. Moreover, SECE produced spatial embeddings that exhibited clearer patterns in low-dimensional visualizations and facilitated a more accurate trajectory inference.

## 1. Introduction

Spatial transcriptomics (ST) captures gene expression profiles with spatial information, providing novel insights into tissue molecular heterogeneity. Applying ST technology plays a crucial role in identifying cell–cell interactions and signaling pathways within the tissue microenvironment, and has enabled groundbreaking discoveries across fields such as neuroscience [1], developmental biology [2], and cancer biology [3]. A variety of ST platforms have been developed, with varying levels of throughput and resolution. Image-based ST platforms, including STARmap [4], seqFISH [5,6], seqFISH+ [7], MERFISH [8] and FISSEQ [9], provide highly accurate gene expression measurement at single-cell resolution but only for a limited number of targeted genes [10]. On the other hand, sequencing-based ST platforms, such as spatial transcriptomics [11] and its commercial version, 10× Genomics Visium; Slide-seqV2 [12]; HDST [13]; Seq-Scope [14] and Stereo-seq [15], can perform high-throughput sequencing on a genome-wide scale with increasing spatial resolution. The spatial resolution of sequencing-based technology continues to improve, with Seq-Scope and Stereo-seq capable of merging subcellular spots based on cellular location to achieve single-cell resolution.

In ST data, spatial domains refer to regions exhibiting consistent patterns in both gene expression and physical location, each with specific anatomical structures [1,16]. Accurately identifying spatial domains is crucial for various downstream analyses, including trajectory inference, cell type deconvolution and cell–cell communications, as well as their biological interpretation. Spatial domains are distinct from cell types, which have been extensively studied in single-cell data. Cell types can be obtained by clustering transcriptional information, and their spatial distribution patterns are uncertain. They may be spatially concentrated, as in the case of excitatory neurons, or discretely distributed, as in the case of astrocytes [17]. Spatial domains are continuous in space, so relying solely on gene expression is insufficient to capture them. It is critical to incorporate spatial information while accurately capturing expression information; this presents a new challenge.

Several methods have been developed to address this challenge. Many existing techniques utilize spatial location information to find neighboring spots for each spot and enhance the similarity between neighbors to ensure the spatial continuity of the domain. Among them, BayesSpace [18] and BASS [19] perform latent variable modeling of regional labels and use Bayesian methods for inference. SpaGCN [20] and STAGATE [21] employ graph convolutional networks and graph attention networks to aggregate neighbor information, respectively. GraphST [22], SpaceFlow [23] and conST [24] utilize self-supervised graph-embedding learning strategies. However, spatial adjacency relationships only represent local information, neglecting to consider the fact that global structure and patterns may result in a lack of comprehensive understanding of the data. In contrast, SpatialPCA [25] introduces a spatially aware dimension reduction method, one which leverages global spatial relationship by measuring the similarity of pairwise spots. Nonetheless, focusing solely on global similarities may overlook subtle spatial details. Moreover, the simultaneous capture of both local and global information, harnessing the advantages of each, remains an area needing further exploration. In addition, some existing methods have limited effectiveness in extracting gene expression features. They use scaled values of highly variable genes (HVG) (e.g., GraphST, SpatialPCA, STAGATE and SpaceFlow) or employ dimensionality reduction techniques like principal component analysis (PCA) (e.g., BASS, SpaGCN, BayesSpace and conST) for expression features. However, these features encounter difficulty in handling expression with high noise, a condition which is very common in barcode-based sequencing ST methods, particularly in high-resolution techniques such as Stereo-seq.

To this end, we developed SECE, an accurate method for identifying spatial domains. SECE first employs an autoencoder (AE) with statistical modeling to obtain gene expression features, which we call cell type-related embedding (CE). Then, it incorporates global and local spatial proximity with CE to learn spatial embedding (SE). Global proximity is quantified by physical distance, and it is thought that the pairwise similarity between spots decreases with longer spatial distances. Local proximity, on the other hand, is determined by expression similarity, aggregating neighbor information based on the similarity of gene expression between spots. SECE utilizes graph attention network (GAT) for SE learning; it aggregates local expression similarity through an attention mechanism while simultaneously constraining the global spatial similarity, using a Gaussian kernel function. Subsequently, by performing clustering on the SE, we can derive the spatial domain to which each spot belongs. SECE also facilitate downstream analyses like visualization and trajectory inference. We demonstrated SECE’s versatility across diverse ST platforms, including high-resolution methods like STARmap/Slide-seqV2/Stereo-seq and lower-resolution platforms like Visium. SECE’s accurate spatial representations in brain and tumor datasets highlight its ability to gain biological insights from complex ST data.

## 2. Materials and Methods

### 2.1. Architecture Overview

SECE is a versatile instrument for modeling ST data across resolutions, including subcellular, single-cell, near-single-cell and multicellular (Figure 1A). It takes the gene expression matrix and spatial coordinate matrix of ST data as input, and outputs spatial domains and embeddings for each spot. First, SECE uses an AE module with a count distribution assumption to compress the expression matrix into expression features. Next, it converts spatial coordinates into local and global position relationships, storing them in the adjacency matrix (ADM) and spatial similarity matrix (SSM), respectively. Then, the GAT module is utilized to balance the expression similarity of local neighbors and the global spatial similarity to obtain the SE (Figure 1B). Finally, spatial domains are identified by clustering SE using mclust [26]. Additionally, downstream analyses, including low-dimensional visualizations [27] and trajectory inference [28,29], are derived from the SE (Figure 1C).

### 2.2. Extracting Expression Features with AE Module

Given ST data with *N* spots and *G* genes, the dimensions of gene expression matrix *X* and spatial coordinate matrix *Y* are *N* × *G* and *N* × 2, respectively. The raw counts *X* are normalized by library size and then log-transformed to obtain the normalized expression matrix $\tilde{X}$. We first employ AE with zero-inflated negative binomial (*ZINB*) or negative binomial (*NB*) distribution [30,31] to compress $\tilde{X}$ into low-dimensional features *Z*. Let *x_ng_* denote the count value of gene *g* in spot *n*; the likelihood function of *x_ng_* under *ZINB* and *NB* distributions is given by
(1)$$ZINB\left(x_{ng};\pi_{ng},\;r_{ng},\;p_{g}\right)=\pi_{ng}\delta_{0}\left(x_{ng}\right)+\left(1-\pi_{ng}\right)\frac{\Gamma \left(x_{ng}+r_{ng}\right)}{x_{ng}!\Gamma \left(r_{ng}\right)}{p_{g}}^{r_{ng}}{\left(1-p_{g}\right)}^{x_{ng}}$$
and
(2)$$NB\left(x_{ng};r_{ng},p_{g}\right)=\frac{\Gamma \left(x_{ng}+r_{ng}\right)}{x_{ng}!\Gamma \left(r_{ng}\right)}{p_{g}}^{r_{ng}}{\left(1-p_{g}\right)}^{x_{ng}},$$
where $\delta_{0}$ is the Dirac delta function, $\pi_{ng}$ is the probability of true gene expression being 0, Γ is the gamma function and (*r_ng_*, *p_g_*) is the standard parametrization for the *NB* distribution. 

The AE module takes $\tilde{X}$ as input, and outputs distribution parameters. The formulation is
(3)$$Z=f_{e}\left(\tilde{X}\right)$$
(4)$$Z^{\prime }=f_{d1}\left(Z\right)$$
(5)$$N\left(\Pi ,R,P\right)=f_{d2}\left(Z^{\prime }\right),$$
where *f_e_* represents an encoder, while *f_d_*_1_ and *f_d_*_2_ constitute the decoders. Specifically, *f_e_* consists of two nonlinear layers, each utilizing Rectified Linear Unit (*ReLU*) activation functions. These layers reduce the feature dimension *G* into *m*′ and *m*, respectively, yielding the expression feature *Z*. Subsequently, *f_d_*_1_ decodes *Z* into *Z*′ with a feature dimension of *m*′. The expression *f_d_*_2_ comprises three output layers which take *Z*′ as input and output three parameter matrices $\left(\Pi ,R,P\right)$ of the *ZINB* distribution, each consisting of elements $\left(\pi_{ng},r_{ng},p_{g}\right)$. The activation functions of these three output layers are exponential, sigmoid and exponential functions, respectively. The expression *f_d_*_2_ employs two layers to learn (*R*, *P*) under the *NB* distribution assumption.

The goal is to minimize the reconstructed loss by minimizing the negative log-likelihood (*NLL*) function, that is, ${{Loss}_{pre}=NLL}_{ZINB}\left(X;\Pi ,R,P\right)$ or *Loss_pre_* = *NllNB*(X; *R*, *P*). We employed *ZINB* for highly sparse data like Stereo-seq and Slide-seq, and *NB* for less sparse data, including STARmap and Visium. Both are implemented in the SECE package.

### 2.3. Capturing Local and Global Relationships

To incorporate physical location information, we construct an ADM and SSM from the spatial coordinates *Y*. ADM summarizes local spatial relationships by storing neighbors for each spot. This local neighborhood information is later employed to adaptively aggregate features in a GAT module based on expression similarity. In contrast, SSM captures global spatial proximity by providing a spatial similarity measure for all pairs of spots, not just neighbors. The SSM is subsequently utilized to constrain the global similarity of SE.

ADM *A* is a *N* × *N*-dimensional symmetric matrix where elements are assigned values of 1 or 0 to indicate neighboring spots or non-neighboring spots, respectively. More precisely, the element *A_ij_* denotes the adjacent relationship between spot *i* and spot *j*:(6)$$A_{ij}=\left\{\begin{array}{c} 0,\;\;\;\;\;\;\;\;\;\;\;\;\;\;\;v_{i}\in N\left(j\right)\;\;\;\;\;\;\;\;\\ 1,\;\;\;\;\;\;\;\;\;\;\;\;\;\;\;v_{i}\notin N\left(j\right).\;\;\;\;\;\; \end{array}\right.$$

Here, $N\left(j\right)$ represents the set of spatial neighbors of spot *j*, which can be determined based on coordinates *Y* by employing K-Nearest Neighbor (KNN) or applying a distance cutoff. By default, we utilize KNN, with the number of neighbors set to 6 for Visium datasets and 10 for other datasets.

SSM Σ is also a *N* × *N*-dimensional symmetric matrix, wherein elements decrease as the distance between spots increases, exhibiting an exponential decay tendency. For spot *i* and spot *j* with coordinates $y_{i}=(y_{i}^{1},\;y_{i}^{2})$ and $y_{j}=(y_{j}^{1},\;y_{j}^{2})$, the corresponding element is:(7)$$\Sigma_{ij}=exp\left(-\frac{{\left\|y_{i}-y_{j}\right\|}_{2}^{2}}{\gamma }\right)$$
where the bandwidth parameter γ controls the spatial influence. By default, γ is set as the 0.05 quantile distance. A larger γ results in a greater spatial influence.

### 2.4. Learning SE with GAT Module

After capturing the expression features and extracting local and global position relationships, we employ the GAT module to learn SE, subsequently clustering SE to delineate spatial domains. The GAT module consists of two GAT layers. 

We first introduce the GAT layer. It takes feature matrix and ADM as inputs, and outputs a new feature matrix after aggregating neighbor information. Let *H* = (*h*_1_, *h*_2_, …, *h_N_*) denote the input feature matrix, which has dimensions *N* × *m*_0_, with *N* samples and *m*_0_ features. The output of GAT layer is denoted as $H^{\prime }=(h_{1}^{\prime },\;h_{2}^{\prime },\;\ldots ,\;h_{N}^{\prime })$ with dimensions $N\times m_{0}^{\prime }$. The GAT layer performs aggregation for each sample adaptively based on the normalized attention scores. For sample *j*, the output feature $h_{j}^{\prime }$ can be formulated as follows:(8)$$h_{j}^{\prime }=\sigma \left(\sum_{i\in N\left(j\right)}\alpha_{ij}Wh_{i}\right)$$
where *W* is a weight matrix with dimensions $m_{0}^{\prime }\times m_{0}$, $N\left(j\right)$ represents the set of neighboring samples of sample *j* and *α_ij_* is the normalized attention coefficient matrix using the SoftMax function: (9)$$\alpha_{ij}={softmax}_{i}\left(e_{ij}\right)=\frac{\exp\left(e_{ij}\right)}{\sum_{k\in N\left(j\right)}\exp\left(e_{kj}\right)}$$
where $e_{ij}=a^{T}\left(Wh_{i}∥Wh_{j}\right),$
*a* is learnable vector and ∥ is the concatenation operation. We used the Exponential Linear Unit (ELU) as activation function σ in the GAT layer.

The GAT module in SECE consists of two GAT layers. It takes expression features *Z* and ADM *A* as input, and outputs SE matrix *U*, which is a *N* × *m*-dimensional matrix:(10)$$U={GAT}_{2}\left({GAT}_{1}\left(Z,A\right),A\right)$$

During neighbor information aggregation using GAT, local expression similarities are captured via attention and adaptively aggregated. To preserve as much information as possible in expression features, the local learning target is the reconstruction loss *L_local_* = *MSE*(*U*, *Z*). We further constrain pairwise correlation using SSM, which including global information, that is, the correlation of each SE at *N* positions *UU^T^* is close to Σ, $L_{global}=MSE\left(UU^{T},\Sigma \right).$ The objective function balances the two similarities by $\lambda_{global}$ and $\lambda_{local}$: (11)$$Loss=\lambda_{global}*L_{global}+\lambda_{local}*L_{local}$$

We kept $\lambda_{local}=1$ unchanged, and adjusted $\lambda_{global}$ based on the ST platform. After obtaining the final SE *U*, we utilized the clustering method mclust [26] to cluster the *U* and determine the spatial domain for each spot.

### 2.5. Architecture of SECE

In the AE module, the dimension of the hidden layer *m*′ and bottleneck layer *m* are 128 and 32, respectively. An Adaptive Moment Estimation (Adam) optimizer is used to minimize *Loss_pre_*, with a learning rate of 1 × 10^−3^ and dropout rate of 0.1. For the GAT module, the dimensions of the two GAT layers are both 32. The Adam optimizer is employed to minimize *Loss*, with a learning rate of 1 × 10^−2^ and dropout rate of 0.2. The default number of iterations for the AE and GAT modules are set to 40 and 50, respectively. 

The hyperparameters $\lambda_{global}$ and $\lambda_{local}$ control the contributions of global and local similarities, respectively, and a larger $\lambda_{global}$ gives greater global influence. Each ST platform has different resolutions and fields of view. For example, spots of ST arrays contain dozens of cells, while Stereo-seq only contains a single cell. Stereo-seq can sequence the hemibrain, while STARmap can only detect a minor region of the visual cortex. We choose a smaller $\lambda_{global}$ value for platforms with more spots and higher resolution. Specifically, for Stereo-seq and Slide-seqV2 data including over 10,000 spots, and with approximately single-cell resolution, we used $\lambda_{global}=0.08$. For Visium data with several thousand spots, $\lambda_{global}$ was set to 0.3. For STARmap data with only 1207 cells, we set $\lambda_{global}$ to 2. For datasets from other platforms, users could also select $\lambda_{global}$ while following this standard.

### 2.6. Datasets

A mouse visual cortex [4] was generated from a STARmap platform with 1207 cells, 1020 genes and a sparsity of 76.88%. STARmap is an in situ sequencing-based ST method with single-molecule resolution. Despite its low gene throughput (160 to 1020 genes), it offers high sensitivity at single-cell resolution, with high efficiency and reproducibility. A mouse hippocampus dataset was generated from Slide-seqV2 [12] platform, with 53,208 cells and 23,264 genes from hippocampus, cortex and thalamus, boasting a high sparsity of 98.19%. Slide-seqV2 offers transcriptome-wide sequencing with near-cellular resolution (10 μm). A mouse olfactory bulb [32] was generated from the Stereo-seq [15] platform, comprising 19,527 cells and 27,106 genes, while 98.69% of values were zero. Stereo-seq is an emerging technique for ST with genome-wide throughput and subcellular resolution. This method captures the expression profile and spatial coordination of each DNA nanoball (DNB) and employs image-based cell segmentation to segment single cells. A mouse hemibrain was also generated from Stereo-seq [15], one which contained 50,140 cells and 25,879 genes, with 96.94% of the values being 0. Human breast cancer data was generated from the Visium platform, containing 3798 spots and 24,923 genes, of which only 77.44% were zero values. Visium is the commercial version of Spatial transcriptomics [11], with a low resolution of 55 μm spots and 1–10 cells per spot [33]. A human dorsolateral prefrontal cortex (DLPFC) [1] dataset was also generated from the Visium platform. In total, 12 DLPFC slices were annotated manually and used as the ground truth of the spatial domain identification. 

For Stereo-seq datasets, cells with an expression level below 200 were removed, according to procedures used in the original studies [15]; then, we filtered the genes expressed in fewer than 20 cells. For other datasets, we screened spots with an expression level below 20 and genes that expressed less than 20 spots. The filtered expression matrix and its corresponding coordinates were input into SECE for analysis. 

For datasets with manual annotation, such as STARmap cortex, DLPFC and breast cancer data, we select the number of domains according to their original study. For mouse hippocampus, hemibrain and olfactory bulb data, we determined the numbers based on the ABA organizational structure.

### 2.7. Evaluation Metrics

The *ARI* evaluates the degree of overlap between the two divisions, which is formulated as
(12)$$ARI=\frac{\sum_{ij}\left(\begin{array}{c} n_{ij}\\ 2 \end{array}\right)-[\sum_{i}\left(\begin{array}{c} a_{i}\\ 2 \end{array}\right)\sum_{j}\left(\begin{array}{c} b_{j}\\ 2 \end{array}\right)]/\left(\begin{array}{c} N\\ 2 \end{array}\right)}{\frac{1}{2}[\sum_{i}\left(\begin{array}{c} a_{i}\\ 2 \end{array}\right)+\sum_{j}\left(\begin{array}{c} b_{j}\\ 2 \end{array}\right)]-[\sum_{i}\left(\begin{array}{c} a_{i}\\ 2 \end{array}\right)\sum_{j}\left(\begin{array}{c} b_{j}\\ 2 \end{array}\right)]/\left(\begin{array}{c} N\\ 2 \end{array}\right)},$$
where *N* is the number of samples and *n_ij_*, *a_i_* and *b_j_* are values from the contingency table. Specifically, *a_i_* represents the number of samples with the real category label *i*; *b_j_* represents the number of samples with the predicted label *j*; and *n_ij_* represents the number of samples with the real category label *i* and the predicted label *j*. In this paper, we utilize *ARI* to evaluate the consistency of the spatial domains identified by various methods with the ground truth domains. The *ARI* ranges from −1 to 1; a greater value indicates better agreement with the true labels.

The *ACC* evaluates the correctness of categories, which is calculated as
(13)$$ACC=\frac{\sum_{i=1}^{N}\delta (r_{i},\;o(s_{i}))}{N},$$
where *N* is the number of samples, and *r_i_* and *s_i_* are the true and predicted spatial domain label of spot *i*. δ is a function that can be defined as
(14)$$\delta \left(x,y\right)=\left\{\begin{array}{c} 1,\;\;\;\;\;\;\;\;\;\;\;\;\;\;\;x=y\\ 0,\;\;\;\;\;\;\;\;\;\;\;\;\;\;\;x\neq y \end{array}\right.$$

*o* is a mapping function that takes the real label *r_i_* as the reference label and then rearranges *s_i_* in the same arrangement, which is implemented using the classical Kuhn–Munkres algorithm [34]. The *ACC* ranges from 0 to 1, and a greater value indicates better performance.

The ASW describes the degree of match between features and category labels. For every spot *i*, silhouette width *S*(*i*) is calculated as
(15)$$S\left(i\right)=\frac{b\left(i\right)-a(i)}{max\{a\left(i\right),b(i)\}}$$
where *a*(*i*) is the average distance between *i* and the points in its cluster, and *b*(*i*) is the lowest average distance from *i* to points in other clusters. In this paper, we use ASW to evaluate how well the SE obtained by various methods explain the known spatial layers. Distance is calculated by SE of various methods, and clusters are annotated spatial layers. The ASW ranges from −1 to 1. A greater ASW indicates better SE learning.

The *LISI* [35] measures the degree of local mixing to evaluate the level of spatial aggregation patterns. For each spot *i*, *LISI* can be formulated as
(16)$$LISI\left(i\right)=\frac{1}{\sum_{l\in L}p_{i}\left(l\right)}$$
where *p*(*l*) is the probability that the spatial domain cluster label *l* exists in the local neighborhood of spot *i*, and *L* is the set of spatial domains. In this paper, we use *LISI* to evaluate the spatial aggregation degree of spatial domains. Local neighborhoods are generated by spatial location, and cluster labels are those predicted by each algorithm. *LISI* values ranges from 0 to 1. A smaller *LISI* indicates better spatial aggregation patterns, i.e., less mixing of cluster labels within local spatial neighborhoods.

### 2.8. Methods for Comparison

We compared SECE with the existing spatial domain identification methods: (1) BayesSpace, implemented in the R package *BayesSpace* V1.4.1 downloaded from https://github.com/edward130603/BayesSpace (accessed on 23 April 2022); (2) SpaGCN, implemented in the Python package *SpaGCN V1.2.2* downloaded from https://github.com/jianhuupenn/SpaGCN (accessed on 23 April 2022); (3) STAGATE, implemented in the Python package *STAGATE_pyG* V1.0.0 downloaded from https://github.com/QIFEIDKN/STAGATE_pyG (accessed on 23 April 2022); (4) BASS, implemented in the R package *BASS* V1.3.1 from https://github.com/zhengli09/BASS (accessed on 3 June 2023); (5) SpaceFlow, implemented in the Python package *SpaceFlow* V1.0.4 from https://github.com/hongleir/SpaceFlow (accessed on 3 June 2023); (6) GraphST, implemented in the Python package *GraphST* V1.4.1 from https://github.com/JinmiaoChenLab/GraphST (accessed on 3 June 2023); (7) SpatialPCA, implemented in the R package *SpatialPCA* V1.2.0 from https://github.com/shangll123/SpatialPCA (accessed on 3 June 2023); and (8) conST (we referred to https://github.com/ys-zong/conST, accessed on 3 June 2023, to run conST). 

We ran each method with its default parameters. For methods that can output spatial features, we used their default feature dimensions. Specifically, the dimensions of GraphST, SpatialPCA, STAGATE and SpaceFlow were 20, 20, 30 and 50 dimensions, respectively. These SEs were used to compute ASW, generate UMAP low-dimensional visualization, and perform trajectory inference. UMAP and domain-level trajectory inference PAGA [28] were computed using ‘scanpy.tl.umap’ and ‘scanpy.tl.paga’ from the scanpy V1.9.3 package, respectively, while cell-level trajectory inference followed Monocle3 V1.0.0 [29] guidelines https://cole-trapnell-lab.github.io/monocle3/ (accessed on 14 May 2022).

## 3. Results

### 3.1. Application to STARmap Data

We first tested SECE on the mouse visual cortex data generated by STARmap [4], with ground truth layers. The 1207 cells were divided into seven layers, including Layer (L)1, L2/3, L4, L5 and L6, as well as the corpus callosum (CC) and hippocampus (HPC) (Figure 2A).

First, we compared the spatial domain identification results of SECE with eight existing methods, including BASS, SpaceFlow, GraphST, STAGATE, SpatialPCA, SpaGCN, BayesSpace and conST (Figure 2B,C). The consistency values for spatial domains and ground truth layers were evaluated using the Adjusted Rand Index (ARI) and Accuracy (ACC) based on Kuhn–Munkres [34]. SECE achieved the highest consistency, with an *ARI* value of 0.65 and an ACC value of 0.79 (Figure 2D). It was closely followed by BASS, with *ARI* of 0.63, however, BASS incorrectly combined HPC and L5 (domain 5), which prevents the utilization of the Kuhn–Munkres algorithm to reassign clusters and compute the accuracy (ACC). Furthermore, other algorithms also exhibited weak spatial aggregation, as indicated by the local inverse Simpson’s index (*LISI*) [35] (Appendix A). Additionally, these methods mistakenly classified aggregated endothelial cells (Endo) in L1 and L2/3 into the same domain, as seen in domain 2 in SpaceFlow, domain 3 in STAGATE and BayesSpace, domain 5 in GraphST and SpatialPCA and domain 6 in SpaGCN (Figure 2C and Appendix A).

Next, we compared the SE of SECE with those of SpaceFlow, GraphST, STAGATE and SpatialPCA. Due to the difficulty encountered by conST in generating clear spatial domains, its SE learning comparison was excluded. The SE of SECE explained the ground truth layers most effectively (ASW = 0.16), followed by SpaceFlow (ASW = 0.12), GraphST (ASW = 0.07), STAGATE (ASW = 0.07) and SpatialPCA (ASW = 0.05) (Figure 2D, right). Moreover, SECE had a clearer and continuous pattern in UMAP visualization, compared to the other methods (Figure 2E). When comparing trajectory inference results, we selected the cortex part in ground truth, namely, L1, L2/3, L4, L5 and L6. The PAGA [28] analysis based on SECE revealed a linear and continuous relationship between these layers (Figure 2F). Conversely, SpaceFlow mistakenly made the connection between L1 versus L4 and L5, while the patterns in the other three methods were more unclear. For individual cells, SECE exhibited a sequential increase of pseudo-time from L6 to L1 (Appendix A) based on Monocle3 [29], and a similar trend was also observed in SpaceFlow. However, GraphST, STAGATE and SpatialPCA ordered L1 and L2/3 incorrectly (Appendix A).

### 3.2. Application to Slide-seqV2 Data

We further tested SECE on the hippocampal dataset generated by the Slide-seqV2 platform [12]. SECE identified 14 distinct domains, and we annotated them according to the known structure of Allen Brain Atlas (ABA) (Figure 3A). The domains were hippocampus (Cornu Ammonis (CA)1, CA2, CA3, Dentate gyrus (DG) and CA slm/so/sr); cortex (Layers 4, 5a, 5b and 6); third ventricle; CC; and three subregions of the thalamus (Figure 3B, left). We further meticulously verified the subtle spatial domains. For example, the four important components of the hippocampal region, CA1, CA2, CA3 and DG, were clearly demarcated, as evidenced by the high expression of their known markers *Wsf1*, *Rgs14*, *Nptxr* and *C1ql2* [36], respectively (Figure 3C and Appendix A). Layer 5 in cortical regions was identified as two sublayers, Layer 5a and 5b, with different gene expression levels (Appendix A). The composition of cell types in each domain further supported the delineation (Appendix A). 

For comparison, we evaluated the performance of existing methods for spatial domain identification (Figure 3D). Notably, SECE was the only approach that could accurately detect subregions in both the cortex and CA. Specifically, for cortical areas, BASS failed to distinguish between Layer 4 and Layer 5, SpaceFlow mixed Layer 5 and Layer 6, SpatialPCA exhibited suboptimal division smoothness, and the remaining algorithms were unable to generate clear cortical regions. In the case of CA, except for GraphST, none of the methods succeeded in identifying the CA2 region. Furthermore, SECE exhibited the highest spatial aggregation performance, as it had the smallest *LISI* values (Appendix A).

We also compared the performance of SE. UMAP generated from SECE clearly displayed clustering patterns for the hippocampus, cortex, thalamus and third ventricle, as well as their subregions (Figure 3B, right). In addition, their sublayers, like Layer 4 and Layer 6, were arranged in a sequence, while UMAP of STAGATE and GraphST missed them (Appendix A). Moreover, we conducted trajectory inference for the cortex due to its continuous relationship between sublayers. We selected the domains corresponding to the cortex in each method and started with the deepest clusters, such as Layer 6 for SECE, domain 1 for SpaceFlow, domain 2 for GraphST and domain 3 for SpatialPCA and STAGATE (Figure 3E,F). For SECE, the pseudo-time consistently increased with decreasing cortical depth, clearly capturing the pseudo-time relationship between different layers. In contrast, SpaceFlow exhibited fewer distinguishable differences between the learned layers. SpatialPCA reversed the relationship between Layer 5b and Layer 6, while GraphST and STAGATE failed to identify the relationships within the cortical region.

### 3.3. Application to Stereo-Seq Data

In this section, we assessed on the mouse olfactory bulb [32] Stereo-seq [15] data. Spatial domains identified by SECE were annotated based on known olfactory bulb layers, including rostral migratory stream (RMS), granule cell layer (GCL), inner plexiform layer (IPL), mitral cell layer (MCL), external plexiform layer (EPL), glomerular layer (GL) and olfactory nerve layer (ONL), from inside to outside (Figure 4A and Appendix A). These domains were validated using known markers [36] for each layer (Figure 4B). Notably, besides the known seven layers, we made a finer division of GCL and ONL, and the sublayers were named GCL-Inner, GCL-Outer, ONL-Inner and ONL-Outer, respectively. Several points of evidence confirmed these sublayers. First, marker gene expression differed between them; specifically, the GCL-inner highly expressed *Nrgn* and the GCL-outer highly expressed *Pcp4*. Markers in the ONL-outer also exhibited higher expression levels compared to those in the ONL-inner. Second, sublayers exhibited distinct cell type composition (Appendix A–D). The GCL-Outer was almost composed of GC, while GCL-Inner contained a certain amount of Oligo, and the GC subtypes in GCL-Inner and GCL-Outer also differed. Moreover, the ONL-Outer almost exclusively contained OEC, while the ONL-Inner contained a fraction of OEC and more Astro. This supported previous findings that the ONL, as a part of the olfactory bulb blood–brain barrier, had fine internal and external subregions, with different cell types [37]. Our study provided further support for this finding at the spatially resolved single-cell level.

We also compared the performance of SECE to the existing methods (Figure 4C). BASS failed to separate the RMS from the GCL layer and could not distinguish GL from EPL. It also identified some irrelevant domains with few cells, such as domains 5, 8 and 9. STAGATE and SpaGCN mixed GL and EPL. SpaceFlow, GraphST and SpatialPCA encountered challenges in accurately dividing the GCL layer. BayesSpace and conST did not yield clear domain identifications. Additionally, BASS had the highest spatial aggregation performance, as well as the lowest *LISI* value, while SECE ranked second (Appendix A). Furthermore, SECE showed robustness when tested with different number of clusters (7, 8 and 10), consistently providing well-bounded ring stratification (Appendix A). Moreover, we assessed the SE learning capabilities. The UMAP visualization based on SECE exhibited a continuous low-dimensional pattern, with nine layers arranged in order of spatial position from inside to outside (Figure 4D). The trajectory inference for these layers demonstrated an approximately linear relationship (Figure 4E). SpatialPCA also displayed linear patterns, except for domains 6 and 7, while the results of SpaceFlow, GraphST and STAGATE exhibited many false positive connections between domains. 

We further applied SECE to a mouse hemibrain dataset with a more complex anatomic structure [15]. SECE achieved spatial domain annotations that were in highly consistent with ABA anatomy (Figure 5A,B(left)). We could clearly separate different regions, including cortical regions, hippocampal regions, midbrain regions, thalamic regions and fiber tracts (FT). The cortex included five layers, L1, L2/3, L4, L5, L6, LVC and CAA, which were supported by their cell type composition (Appendix A). In contrast, other methods failed to accurately identify these cortical layers (Figure 5C and Appendix A). Specifically, all of them encountered difficulties in identifying Cortex L5 with high smoothness. In addition, *LISI* showed that spatial domains identified by SECE had the strongest spatial pattern (Appendix A). Furthermore, low-dimensional visualization of SE using t-SNE [38] showed the effects of aggregation in the same region and separation in different regions (Figure 5B, right).

### 3.4. Application to Visium Data

Finally, we tested the applicability of SECE on Visium dataset. The breast cancer dataset was divided into four phenotypic regions according to pathological images: ductal carcinoma in situ/lobular carcinoma in situ (DCIS/LCIS), healthy tissue (Healthy), invasive ductal carcinoma (IDC) and tumor-surrounding regions with low features of malignancy (Tumor edge), as well as 20 annotated subdivisions [32] (Figure 6A and Appendix A).

We initially segmented 20 domains using each algorithm and performed phenotype annotations, which were then compared with image-based manual annotations (Figure 6B and Appendix A). Notably, in the context of a dataset characterized by a low missing rate of 77.44%, all algorithms demonstrated commendable performance. Most of the algorithms had *ARI* values ranging from 0.55 to 0.62 (Appendix A). We found that SECE divided individual tumor regions into wrapped layers more clearly. Specifically, cluster 20 and cluster 15 divided the IDC-5 into internal and external layers, while cluster 11 and cluster 10 split the DCIS/LCIS-1. (Figure 6B and Appendix A). To investigate the biological significance of the refined stratification, we named clusters 20,15,11 and 10 as IDC-inner, IDC-outer, DCIS/LCIS-inner and DCIS/LCIS-outer, respectively. We also focused on the parts of clusters 4 and 5 that were located near the tumor edge, referred to as IDC-edge and DCIS/LCIS-edge. (Figure 6C). We analyzed the cell type composition of each spot in these domains by integrating the annotated breast cancer scRNA-seq data [39] and deconvoluting each spot using cell2location [40] (Appendix A). The proportion of tumor cells gradually decreased as the edge of the tumor was approached, while those of immune cells and stromal cells gradually increased (Figure 6D), confirming the correctness of the subregions. 

We further explored the characteristics of these subregions (-edge, -outer and -inner) of DCIS/LCIS and IDC (Figure 6E). Differential expression analysis [41] showed that DCIS/LCIS-edge had a high level of humoral immunity, which could be confirmed by the enrichment of B cell receptor signaling pathway-related genes (*IGLC2*, *IGLC3*, *IGHG1*, *IGHG3* and *IGHA1*). The DCIS/LCIS-outer subregion overexpressed *KRT14* of the keratin family, which is a key regulator of metastasis, suggesting invasive potential [42,43]. In contrast, DCIS/LCIS-inner subregion showed non-metastatic traits but also had tumor-promoting abilities, as indicated by a high expression of *LDHA* [44]. As for the IDC subregions, IDC-edge showed elevated expression of biomarkers linked to tumor proliferation, invasion and migration, including tumor-associated macrophage (*APOE*), complement components (*C1qA*, *C1qB*), cathepsin (*CTSD*), and apolipoprotein (*APOC1*) [45,46,47,48,49]. IDC-outer had a high level of immunity and some transferability, derived from the high expression of MHC class I-related genes (*HLA-A*, *HLA-B* and *B2M*) [50,51]. Besides, there was increased expression of genes such as *MGST1* and *MRPS30-DT*, which have been known to promote breast carcinoma cell growth and metastasis [52,53]. In IDC-inner, there were higher levels of tumor activity and lower levels of immune response. *LINC00052*, known to promote breast cancer cell proliferation by increasing signals of epidermal growth factor receptor (EGFR) such as *HER3* [54,55,56], was overexpressed. Upregulation of *COX6C* and *FAM234B* implied higher levels of cellular respiration [57] and lower immune response function [58], respectively.

Furthermore, we inferred developmental trajectories for the IDC and DCIS/LCIS regions. The starting points for calculating pseudo-time in IDC and DCIS/LCIS were the interior of the tumor regions (Figure 6F). There were increased pseudo-time values as the region of the tumor moved outward, effectively mimicking the gradual progression of tumor development.

Moreover, to further evaluate the power of SECE in spatial domain identification, we tested 12 human dorsolateral prefrontal cortex (DLPFC) datasets generated from the Visium platform. The original study [1] had manually annotated the spatial domains of these datasets, encompassing white matter and six cortical layers (Figure 7A and Appendix A). The spatial domain identified by SECE exhibited the highest levels of agreement with the original annotations (Figure 7B,C and Appendix A). The median *ARI* for SECE was 0.58, surpassing the second- and third-ranked algorithms, STAGATE and SpatialPCA, which achieved median *ARI* values of 0.55 and 0.54, respectively (Figure 7D). These findings highlighted the superior performance of SECE in accurately delineating the spatial domains within the low-resolution data.

## 4. Discussion

SECE captures both local and global relationships among spots and aggregates their information using expression similarity and spatial similarity, respectively. This approach enables precise spatial domain division and facilitates interpretable spatial embedding learning across diverse ST datasets. Moreover, the AE module that explicitly models gene expression counts enhances SECE’s ability to handle noisy data. 

With the increases in captured area within the ST data and advancements in resolution, there is a growing demand for computational methods which can be used to exhibit higher efficiency and scalability. We recorded the runtime and GPU memory consumption for each dataset (Appendix A). For Slide-seqV2 hippocampus data and the Stereo-seq hemibrain data, which contained over 50,000 cells, SECE achieved a running time of less than 4.2 min while utilizing less than 5GB of GPU memory. These results demonstrated the superior computational efficiency and scalability of SECE when dealing with large-scale datasets.

While SECE has demonstrated notable performance, there are still several aspects that can be further enhanced. Firstly, we employ a pre-defined SSM to characterize global similarity, but exploring more flexible global correlation patterns could be advantageous. Secondly, we only utilized ST data as inputs, but incorporating matching histology data may provide additional benefits [20]. Although matching histology image data is currently only available on specific platforms like Visium, we can still leverage such images as optional Appendix A when available. Finally, integrating single-cell data with ST data can enhance the data quality of the latter, increasing the throughput, or reducing the noise in the gene expression [59]. Therefore, incorporating single-cell data is another approach which can be used to improve spatial representation capabilities.

## Figures and Tables

**Figure 1 biomolecules-14-00674-f001:**
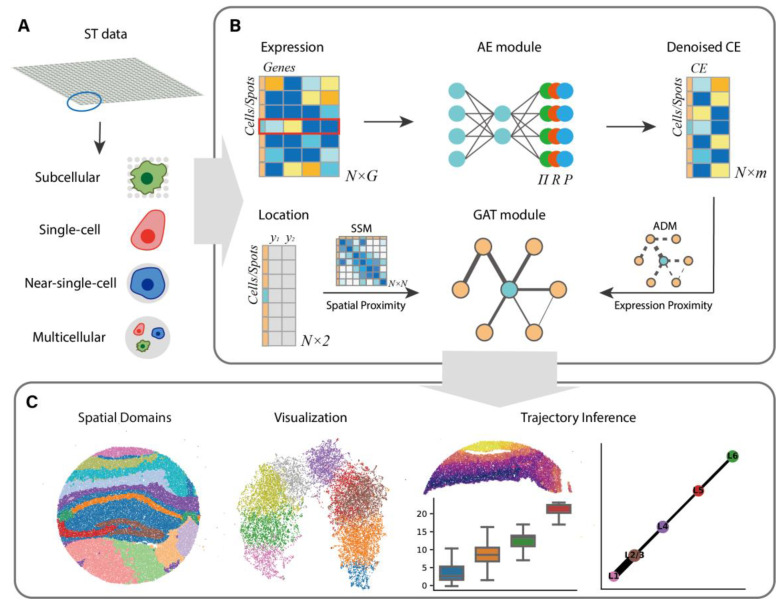
Overview of SECE. (**A**) SECE is applicable to spatial transcriptome (ST) data with different resolutions, including subcellular, single-cell, near-single-cell and multicellular resolutions. (**B**) SECE takes gene expression profiles and spatial coordinates as inputs. It begins with an autoencoder (AE) module that compresses gene expression into low-dimensional features based on a count distribution. Then, a graph attention network (GAT) module learns spatial embeddings (SE) by balancing local expression similarity and global spatial proximity, as measured by the distances between expression features and spatial coordinates, respectively. (**C**) The main SECE outputs include identified spatial domains, low-dimensional visualizations, and inferred spatial trajectories, which are obtained by running clustering, visualization and trajectory inference on the SE.

**Figure 2 biomolecules-14-00674-f002:**
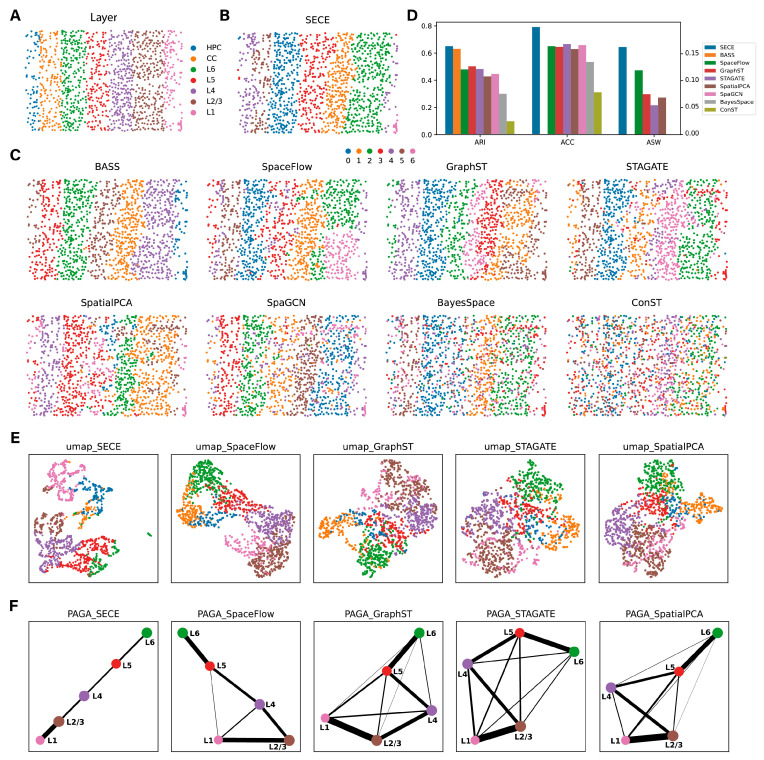
Application of SECE to mouse visual cortex STARmap data. (**A**) Layer structure of the tissue section from the original study. (**B**) Spatial domains identified by SECE. (**C**) Spatial domains identified by BASS, SpaceFlow, GraphST, STAGATE, SpatialPCA, SpaGCN, BayesSpace and conST. (**D**) Assessment of spatial domain identification and SE learning across methods using ARI, NMI and ASW. (**E**) UMAP visualizations generated by SECE, SpaceFlow, GraphST, STAGATE and SpatialPCA, colored by annotated layers. (**F**) PAGA graphs generated by SECE, SpaceFlow, GraphST, STAGATE and SpatialPCA.

**Figure 3 biomolecules-14-00674-f003:**
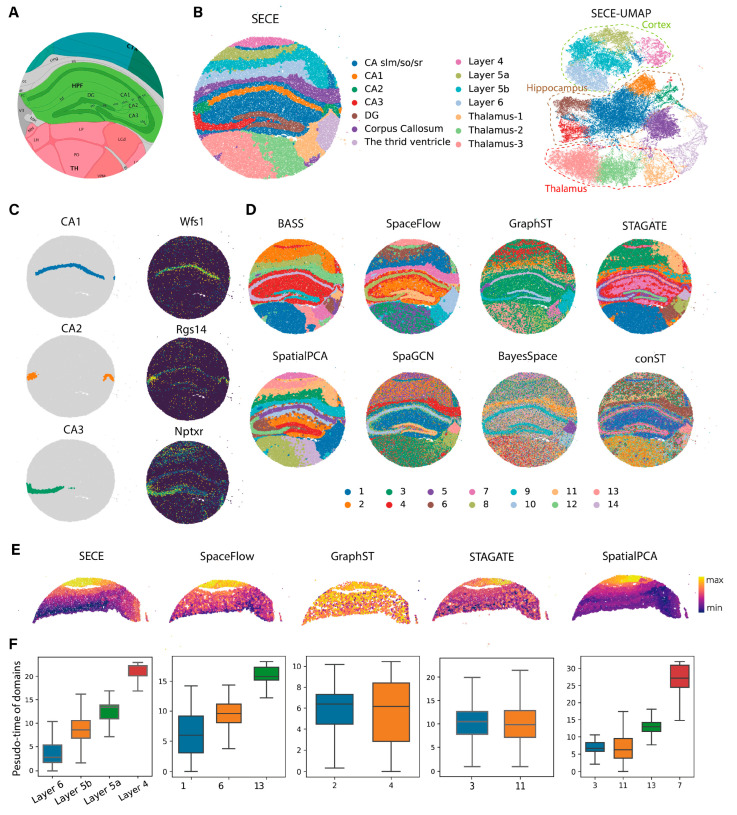
Application of SECE to mouse hippocampus Slide-seqV2 data. (**A**) Annotation of hippocampus structures from the Allen Brain Atlas (ABA) for adult mouse brain. (**B**) Left: spatial visualization of domains identified by SECE. Right: UMAP visualization of domains identified by SECE. Spatial domains were annotated based on the ABA structures. (CA, Ammon’s horn; DG, Dentate gyrus.) (**C**) Spatial visualization of CA1, CA2 and CA3 domains identified by SECE (**Left**) and the corresponding marker genes *Wsf1*, *Rgs14* and *Nptxr* (**right**). (**D**) The 14 spatial regions identified by BASS, SpaceFlow, GraphST, STAGATE, SpatialPCA, SpaGCN, BayesSpace and conST. (**E**) Pseudo-time of each cell, calculated by Monocle3 based on SECE, SpaceFlow, GraphST, STAGATE and SpatialPCA embeddings. (**F**) Pseudo-time of the cells in each isocortex layer based on SECE, SpaceFlow, GraphST, STAGATE and SpatialPCA.

**Figure 4 biomolecules-14-00674-f004:**
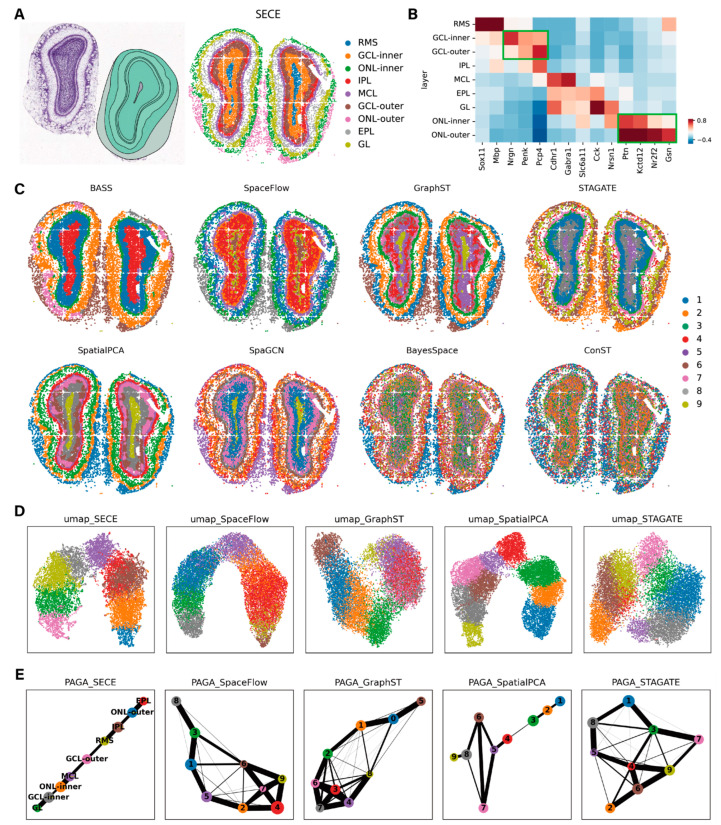
Application of SECE to mouse olfactory bulb Stereo-seq data. (**A**) **Left**: annotation of mouse olfactory bulb structures from the ABA. **Right**: spatial visualization of domains identified by SECE. (RMS, Rostral migratory stream; GCL, Granule cell layer; IPL, Inner plexiform layer; MCL, Mitral cell layer; EPL, External plexiform layer; GL, Glomerular layer; ONL, Olfactory nerve layer.) (**B**) Heatmap of known marker gene expression for each layer; the median values of the centered and standardized gene expression for each region are shown. The green boxes are the subregions of GCL and ONL, respectively. (**C**) The 9 spatial regions identified by BASS, SpaceFlow, GraphST, STAGATE, SpatialPCA, SpaGCN, BayesSpace and conST. (**D**) UMAP visualizations generated by SECE, SpaceFlow, GraphST, STAGATE and SpatialPCA, colored by annotated layers. (**E**) PAGA graphs generated by SECE, SpaceFlow, GraphST, STAGATE and SpatialPCA.

**Figure 5 biomolecules-14-00674-f005:**
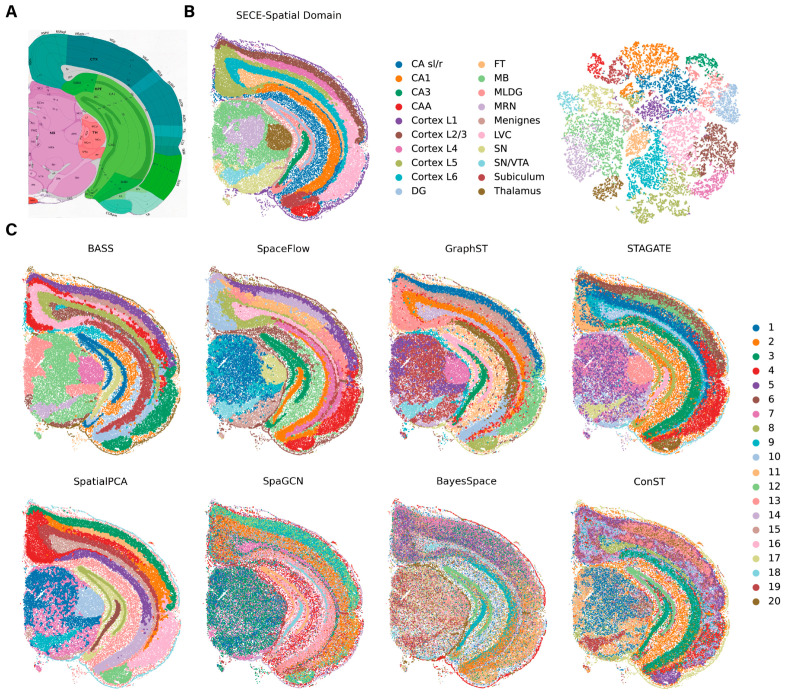
Application of SECE to mouse brain Stereo-seq data. (**A**) The annotation of mouse hemibrain structures from the ABA. (**B**) **Left**: spatial visualization of domains identified by SECE. **Right**: UMAP visualization of domains identified by SECE. Spatial domains were annotated based on ABA. (sl/r, stratum lacunosum/raditum cornu ammonis; DG, dentate gyrus; FT, fiber tract; MLDG, molecular layer of dentate gyrus; MRN, midbrain reticular nucleus; SN, substantia nigra; VTA, ventral tegmental area.) (**C**) The 20 spatial regions identified by BASS, SpaceFlow, GraphST, STAGATE, SpatialPCA, SpaGCN, BayesSpace and conST.

**Figure 6 biomolecules-14-00674-f006:**
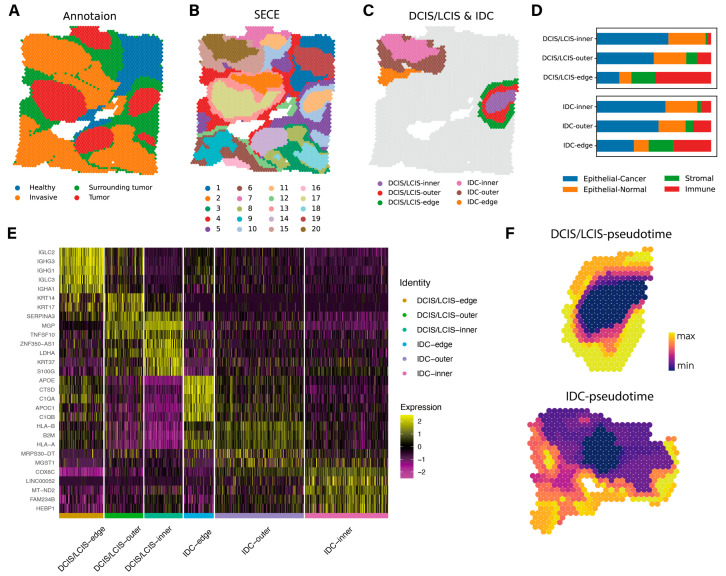
Application of SECE to human breast cancer Visium data. (**A**) Pathology annotation of the tissue section from the original study. (**B**) Spatial regions identified by SECE. (**C**) Fine-grained regions identified by SECE, that is, DCIS/LCIS-inner, DCIS/LCIS-outer, DCIS/LCIS-edge, IDC-inner, IDC-outer and IDC-edge. (**D**) Cell type compositions of 6 fine-grained regions. (**E**) Heatmaps of normalized expression of signature genes identified in the DE analysis based on the 6 fine regions. (**F**) Pseudo-times of spots calculated by Monocle3 in the DCIS/LCIS (top) and IDC regions (bottom).

**Figure 7 biomolecules-14-00674-f007:**
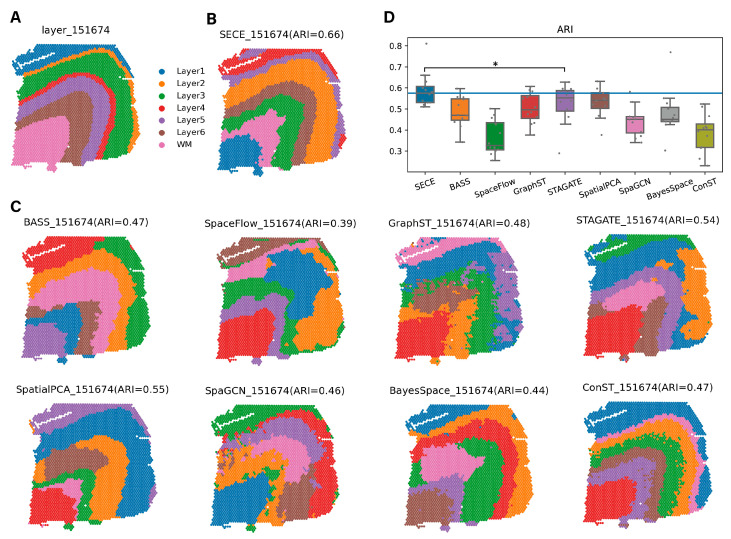
Application of SECE to DLPFC Visium data. (**A**) Pathology annotation of section 151674 from the original study. (**B**) Spatial regions of the section 151674 identified by SECE. (**C**) Spatial regions of the section 151674 identified by eight other methods. (**D**) Boxplot of *ARI* values for 12 slices (*p* = 0.02, one-tailed paired *t* test). * represents the p value less than 0.05. In the boxplot, the center line denotes the median, box limits denote the upper and lower quartiles, and whiskers denote the 1.5× interquartile range. The blue horizontal line indicates the median *ARI* value of SECE.

## Data Availability

Slide-seq datasets are available at https://portals.broadinstitute.org/single_cell/study/slide-seq-study (accessed on 10 April 2022). The Stereo-seq mouse hemibrain dataset is available at https://db.cngb.org/stomics/mosta/ (accessed on 25 April 2022). The Stereo-seq olfactory bulb dataset is available at https://github.com/JinmiaoChenLab/SEDR_analyses (accessed on 28 March 2022). The Slide-seqV2 hippocampus datasets are available at https://singlecell.broadinstitute.org/ (accessed on 4 May 2022). The STARmap mouse visual cortex dataset is available at http://clarityresourcecenter.org/ (accessed on 4 May 2022). The Visium human breast cancer dataset is available at https://www.10xgenomics.com/resources/datasets/human-breast-cancer-block-a-section-1-1-standard-1-1-0 (accessed on 8 August 2022). The Visium DLPFC dataset is available within the spatialLIBD package (http://spatial.libd.org/spatialLIBD, accessed on 22 June 2023). We also organized these raw data, including expression counts and coordinates, as well as H and E images of Visium, into a format that is easy to read by SCANPY; the results are available at https://drive.google.com/drive/folders/1uHc2F_e1PX1Q_efuO5xrFw9bhJa0wCm4. The SECE algorithm is implemented and provided as a pip installable Python package, which is available on Github https://github.com/yuyuanyuana/SECE. The source code and datasets are available at Zenodo https://zenodo.org/record/8130682.

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
