# Peer review of "Accurate Identification of Spatial Domain by Incorporating Global Spatial Proximity and Local Expression Proximity"

_biomolecules, 2024, doi:10.3390/biom14060674_

Round 1

Reviewer 1 Report

Comments and Suggestions for Authors

SECE offers a versatile and comprehensive framework to analyze ST data. It includes the use of graph attention networks for balancing expression and spatial similarities, as well as the ability to handle data across different resolutions. The authors organized the presentation very well in their paper. I think overall it's ready to go but it has a missing piece that I'd like to invite the authors to supplement. 

DeepST is another deep learning framework published in NAR, 2022 for ST data analysis, which combines spatial location, histology, and gene expression to model spatially embedded representations. The authors of DeepST claimed it outperforms existing methods on benchmark datasets, such as the human dorsolateral prefrontal cortex. I’d recommend the authors to compare the performance of SECE with DeepST. It would benefit the community for the best practice and further improving the methodology in the field.

Reviewer 2 Report

Comments and Suggestions for Authors

SECE: accurate identification of spatial domain by incorporating global spatial proximity and local expression proximity

Major comments:

1.       It is unclear what does SECE mean? In Figure 1 it shows CE as reduced dimensions but it was not explained in the text.

2.       Eq. (3) and (4) are identical.

3.       Eq. (7) needs to be rewritten for y being a two-dimensional vector, not a scalar.

4.       In Eq. (8) the dimension of h_j^prime is m0^prime, while in Eq. (10) it says the ouput matrix is a Nxm dimension. It is unclear whether the dimension of the output is m0, m0^prime, or m?

5.       What is the difference between GAT1 and GAT2?

6.       Figure 2E and other similar plots. It is unclear how the UMAPS were generated for each method. What is the dimension of the feature matrix for each method?

7.       Use of mclust is unclear. How to determine the number of clusters?

Reviewer 3 Report

Comments and Suggestions for Authors

In this manuscript, the authors propose a deep-learning based approach to decipher tissue microenvironments and associated biological functionality. The merit of this manuscript encompasses a significant improvement over the existing methods. I have a few comments that the authors should address:

1. The introduction does not currently provide a comprehensive survey of the current shortcomings of other existing methods by peers. It only provides a rather narrow architecture. The manuscript would benefit from a more comprehensive discussion and addressing the pertinent challenges/shortcomings.

2. Are the hyperparameters obtained from the model training for expression and spatial similarity distinct? Is there a model performance issue in case of any degree of overlap?

3. How are the statistical significance evaluated for the pertinent data in Figure 6 and Figure 7? Please elaborate.

Round 2

Reviewer 3 Report

Comments and Suggestions for Authors

The authors have addressed this reviewer's concerns adequately.